# OMA1-Mediated Mitochondrial Dynamics Balance Organellar Homeostasis Upstream of Cellular Stress Responses

**DOI:** 10.3390/ijms25084566

**Published:** 2024-04-22

**Authors:** Robert Gilkerson, Harpreet Kaur, Omar Carrillo, Isaiah Ramos

**Affiliations:** 1School of Integrative Biological & Chemical Sciences, The University of Texas Rio Grande Valley, Edinburg, TX 78539, USA; hkaurchd096@gmail.com (H.K.); agustin_omar@outlook.com (O.C.);; 2Department of Health & Biomedical Sciences, The University of Texas Rio Grande Valley, Edinburg, TX 78539, USA

**Keywords:** mitochondria, fusion, fission, bioenergetics, transmembrane potential, OPA1, OMA1, DRP1, cristae, apoptosis, autophagy, integrated stress response

## Abstract

In response to cellular metabolic and signaling cues, the mitochondrial network employs distinct sets of membrane-shaping factors to dynamically modulate organellar structures through a balance of fission and fusion. While these organellar dynamics mediate mitochondrial structure/function homeostasis, they also directly impact critical cell-wide signaling pathways such as apoptosis, autophagy, and the integrated stress response (ISR). Mitochondrial fission is driven by the recruitment of the cytosolic dynamin-related protein-1 (DRP1), while fusion is carried out by mitofusins 1 and 2 (in the outer membrane) and optic atrophy-1 (OPA1) in the inner membrane. This dynamic balance is highly sensitive to cellular stress; when the transmembrane potential across the inner membrane (Δψ_m_) is lost, fusion-active OPA1 is cleaved by the overlapping activity with m-AAA protease-1 (OMA1 metalloprotease, disrupting mitochondrial fusion and leaving dynamin-related protein-1 (DRP1)-mediated fission unopposed, thus causing the collapse of the mitochondrial network to a fragmented state. OMA1 is a unique regulator of stress-sensitive homeostatic mitochondrial balance, acting as a key upstream sensor capable of priming the cell for apoptosis, autophagy, or ISR signaling cascades. Recent evidence indicates that higher-order macromolecular associations within the mitochondrial inner membrane allow these specialized domains to mediate crucial organellar functionalities.

## 1. Mitochondrial Organization Mediates Organellar Function

Within mammalian cells, mitochondria are maintained as a double membrane-bound organellar network. Mitochondria are bounded by the outer membrane, with the inner membrane organized into lamellar or tubular cristae to maximize the surface-to-area ratio and metabolic capacity. The intermembrane space (IMS) is contained between the two membranes, while the matrix is enclosed within the inner membrane. These membranes and organellar compartments were first revealed by thin-section transmission electron microscopy [1,2], while more recent studies using electron tomography have revealed that cristae may be either tubular or lamellar and are contiguous with the inner membrane at discrete cristal junctions [3,4]. These membranes and compartments provide the environment for the mitochondrial proteome, ~1500 proteins that carry out a multitude of mitochondrial biochemical functions, encompassing metabolism and bioenergetics, as well as a variety of signaling functions [5,6,7]. The overwhelming majority of mitochondrial proteins are encoded by nuclear genes, transcribed, translated, and imported into the organelle by the mitochondrial protein import machinery. As an endosymbiontically derived organelle, however, a small number of crucially important factors are encoded by human mitochondrial DNA (mtDNA); this small (16,569 bp), circular DNA encodes 12 polypeptides, 2 rRNAs, and 14 tRNAs that are required for the five complexes of oxidative phosphorylation (OXPHOS) to be assembled and fully functional. Complexes I, II, III, and IV are large, multisubunit protein complexes in the inner membrane, where they carry out electron transfer from NADH and FADH_2_ to oxygen (O_2_), which provides the energy to pump protons (H^+^) from the matrix to the IMS. This proton pumping generates the mitochondrial transmembrane potential (Δψ_m_), the electrochemical gradient that powers the F_1_F_0_ ATP synthase (Complex V); ATP synthase allows protons from the IMS to flow back into the matrix down the gradient, providing the energy to catalyze bond formation between ADP and Pi to generate ATP [8,9]. Complexes I, III, and IV have been shown to form a ‘supercomplex’ or respirasome [10,11,12], indicating that higher-order macromolecular association between inner-membrane complexes is necessary for optimal bioenergetics. Similarly, the F_1_F_0_ ATP synthase functionally interacts as a dimer, allowing for the tubulation of mitochondrial cristae [13,14]. As such, the complexes that mediate mitochondrial bioenergetics are directly and mechanistically connected to the internal structural dynamics of the mitochondrial network.

The elegant, well-regulated interior structure of mitochondria is echoed by their organization within the cell at large. From their earliest characterizations as ‘thread-like granules’, giving rise to their designation as mitochondria, it has become increasingly apparent that mitochondria exist as a highly dynamic network. Skulachev and co-workers first used photobleaching to demonstrate that mitochondria form an interconnected network [15]. Advances in biological imaging revealed that mitochondria exist as a highly dynamic network, balancing between a reticular, interconnected state and a population of fragmented individual organelles [16]. This network organization allows for the efficient distribution of mitochondrial components. As an example, mtDNA is typically present at ~1000 copies per cell, complexed into protein–DNA assemblies called nucleoids. MtDNA nucleoids are arrayed through the mitochondrial network at discrete foci to provide mtDNA-encoded gene products for oxidative phosphorylation [17,18]. Similarly, mitochondrial ribosomes and other components are localized to discrete sites along the network [19]. Even within highly structured cell settings such as skeletal muscle, in which contractile actomyosin filaments dominate the cellular structure, mitochondria retain a highly interconnected organization, both among subsarcolemmal and interfibrillar mitochondrial populations [20]. Moreover, this dynamic organization is also coordinately regulated with mitochondrial biogenesis; murine genetic models lacking the PGC-1α and PGC-1β coactivators responsible for mitochondrial biogenesis also have dysregulated expressions of fusion and fission factors [21]. Both PGC-1α and β are positive regulators of mitofusins [22]. As with the structure of the inner membrane into cristae, the highly dynamic processes of mitochondrial fission and fusion use distinct sets of stress-responsive factors to modulate and adapt mitochondrial organization in response to cellular cues.

## 2. Mitochondrial Fission Is Mediated by Recruitment of Membrane-Shaping Proteins

To distribute the organellar network as a population of individual organelles, mitochondria undergo division, or fission, events driven by dynamin-related protein-1 (DRP1) [23]. As with canonical dynamins, DRP1 is a cytosolic GTPase that is recruited to the mitochondrial outer membrane, where it forms a multimeric collar that constricts to pinch the mitochondrial tubule into two daughter organelles (Figure 1). The recruitment of cytosolic DRP1 to mitochondria is carried out by the coordinated action of endoplasmic reticulum–mitochondrial contacts and actin/myosin active transport, in which actin filaments polymerize in the mitochondrial outer membrane, delivering DRP1 at discrete sites for subsequent fission events [24,25,26]. To assemble at fission sites, DRP1 is bound by a variety of receptor proteins in the outer membrane. Fis1 was the first identified mammalian DRP1-binding partner for mitochondrial fission [27]; however, MFF was subsequently found to play a similar but independent role in binding mitochondrial DRP1 in the outer membrane [28]. MiD49 and MiD51 are additional mitochondrial DRP1-binding factors [29,30], providing multiple recruitment partners for DRP1 in the outer membrane for cooperative recruitment. Upon recruitment to mitochondrial fission sites, DRP1 assembles into a multimeric ring, followed by constriction [23]. Notably, this GTP-dependent activation of DRP1-mediated fission is controlled by the inhibitory phosphorylation of DRP1 (Ser656 in rat); thus, the dephosphorylation of DRP1 is necessary for fission to occur [31,32]. When phosphorylated, mitochondrial DRP1 accumulates into large, stable aggregates that do not carry out fission [33]. A range of post-translational modifications of DRP1 provide additional mechanisms for signaling pathways to modulate DRP1-mediated fission. Voeltz’ group found that Dyn2 then carries out the final steps of mitochondrial scission [34], while alternately, mice lacking Dyn2 are able to carry out mitochondrial fission [35], indicating that there may be context-specific nuances to this step. As mitochondrial organization is a homeostatic balance of fission and fusion, the knockdown or disruption of fission factors DRP1, Dyn2 [34], Fis1 [36], Mff [28], MiD49, or MiD51 [37] elicits a strong shift toward a highly interconnected mitochondrial organization. This is directly connected to the overall well-being of the cell; the genetic ablation of DRP1 in mice leads to muscle atrophy, lack of growth, and death [38], demonstrating the importance of mitochondrial fission for cellular homeostasis and viability.

## 3. OMA1 Links Mitochondrial Fusion with Bioenergetic Function

As the converse process to fission, mitochondrial fusion links discrete individual organelles to establish a continuous mitochondrial tubule. Just as this process is mechanistically distinct from fission, fusion events in the mitochondrial outer and inner membranes are carried out by distinct sets of interacting factors. The fusion of the outer mitochondrial membrane is mediated by the dynamin-related GTPases mitofusin 1 and 2 (MFN1 and MFN2) [39,40]. Mitofusin heptad repeat regions (HR2) allow oligomeric tethering, bringing individual organelles into close proximity [41], while the dimeric association of GTPase domains is necessary for membrane fusion [42]. MFN1 and 2 can also undergo heterodimeric tethering to establish continuity of the outer membrane [43].

Consistent with the highly ordered nature of mitochondrial inner-membrane organization, optic atrophy-1 (OPA1) plays direct mechanistic roles in both fusion of the inner membrane and cristal organization. Located within the inner membrane, OPA1 is oriented towards the inner-membrane space [44], with two full-length long isoforms (L-OPA1) capable of carrying out fusion of the inner membrane [45]. OPA1 accomplishes this by binding homotypically to either itself or to cardiolipin, a mitochondrial-specific lipid species found in the inner membrane [46]. Critically, OPA1-mediated mitochondrial fusion is sensitive to transmembrane potential; under steady-state conditions with an intact Δψ_m_, long (L-OPA1) isoforms carry out inner-membrane fusion, while loss of Δψ_m_ causes the proteolytic cleavage of L-OPA1 to short, fusion-inactive S-OPA1 isoforms, leading to mitochondrial fragmentation (Figure 1) [47,48]. While S-OPA1 isoforms are unable to mediate membrane fusion, recent evidence suggests they have key roles in protection from cellular stress such as that from oxidative radicals [49,50]. In addition to, but distinct from, its role in mediating inner-membrane fusion, OPA1 also helps to maintain the structure of cristae, providing an increased surface-to-area ratio for OXPHOS metabolism. Cristae are formed by the tubulation of the inner membrane, with the cristal junction forming the ~17 nm collar-like interface of the cristal membrane with the inner boundary membrane [51]. Crista junctions are maintained as regulated structures by the mitochondrial contact site and cristae organizing system (MICOS), a macromolecular association composed of the core factors Mic60 and Mic10, as well as additional factors [52]. To form cristae junctions, the MICOS complex associates with OPA1 [53,54,55], while the dimerization of the F_1_F_0_ ATP synthase provides curvature and tubulation of the membrane [13,56]. Within the cristal membrane, the OXPHOS complexes are enriched, relative to the inner boundary membrane, for a highly specialized submitochondrial compartment for maximal ATP production [57]; more recently, improved experimental methods have indicated that individual cristae may have distinct bioenergetic parameters from their neighboring cristae, with the CM and IBM having distinct Δψ_m_ values [58]. Notably, OPA1’s role in crista maintenance is mechanistically distinct from its ability to maintain inner-membrane fusion [53], as OPA1 appears to cooperatively interact with the MICOS complex to maintain crista junctions. The loss or knockdown of OPA1 disrupts crista junctions [44,59]; however, the complete knockout of OPA1 reduces, but does not eliminate, crista junctions, suggesting that OPA1 modulates crista formation, while MICOS is the major determinant [60], with mitochondrial solute carrier SLC25A interacting with OPA1 to mediate this role in crista junction maintenance [61]. OPA1 thus plays two crucial roles in shaping the inner membrane: fusion establishes the continuity of the inner membrane by mediating reticular interconnection of the organelle, and cristal formation, in which the inner membrane is organized into lamellae or tubules for maximal bioenergetic output. Crucially, OPA1’s role in modulating inner-membrane topology is sensitive to cellular stresses; under steady-state conditions, L-OPA1 long isoforms carry out their distinct roles in mediating both mitochondrial inner-membrane fusion and cristal structure. When Δψ_m_ is lost, however, L-OPA1 isoforms are cleaved by the OMA1 metalloprotease [62,63].

OMA1 works in concert with the YME1L inner-membrane protease, which carries out a modest, constitutive level of L-OPA1 processing at OPA1’s S2 site, resulting in the steady-state balance of long and short OPA1 isoforms [64]. Following the loss of Δψ_m_, however, OMA1 is activated, cleaving L-OPA1 at its S1 site. This causes a complete loss of fusion-active L-OPA1, rendering inner-membrane fusion impossible and leaving mitochondrial fission unopposed. As a result, the network fragments to a disconnected collection of vesicular organelles [64,65]. Conversely, the genetic knockout of OMA1 results in mitochondria retaining a fused morphology even during the loss of Δψ_m_ via hydrogen peroxide [64] or uncouplers such as carbonyl cyanide *m*-chlorophenyl hydrazone (CCCP) [65]. Moreover, cells lacking OMA1 display mitochondrial alterations leading to metabolic dysregulation in mouse models [66]. Thus, effective mitochondrial dynamic homeostasis requires intact stress-sensitive OMA1. Following activation and L-OPA1 proteolysis, OMA1 undergoes the self-cleavage and degradation of OMA1 [67,68]. OMA1 is localized to the inner membrane, with a C-terminal proteolytic domain on the IMS face and an N-terminal unstructured domain on the matrix face that is required for activation upon loss of Δψ_m_ [65]. In addition, OMA1 interacts with a variety of factors in the inner membrane, including prohibitin [67,69] and CHCHD2 [70], though the higher-order organization and mechanistic implications remain unclear, as well as the fundamental activation of OMA1 by the loss of Δψ_m_. While these mechanistic questions remain open, OMA1 has clearly emerged as a major stress sensor that controls mitochondrial dynamics and primes the organelle for a range of signaling pathways. OMA1 thus plays a unique role as a mitochondrial stress sensor, capable of collapsing mitochondrial structure/function homeostasis (Figure 1) and promoting cell-wide stress responses.

## 4. OMA1 Impacts on the Cell at Large: Apoptosis, Autophagy, and Integrated Stress Response

The factors dedicated to modulating mitochondrial form are directly and mechanistically linked to a broad range of cell-wide signaling pathways. The discovery that the mitochondrial electron carrier cytochrome c activates apoptotic caspases upon its release from the organelle [71,72] revolutionized mitochondrial roles beyond bioenergetics, elevating the perception of mitochondria from a solely bioenergetic set of batteries to a sensitive network directly connected to the very life or death of the cell. This was followed later by other cellular signaling pathways, including autophagy and the integrated stress response, in which mitochondrial membrane-shaping proteins play key mechanistic roles linking mitochondrial dynamics with cell stress response pathways. Mitochondria continue to emerge as a crucial upstream stress sensing network, with impacts far beyond bioenergetics. In particular, OMA1, as the chief mitochondrial stress sensitive protease, is emerging as a critical checkpoint for cellular homeostasis and health.

Mitochondria participate in the intrinsic apoptotic pathway, in which the pro-apoptotic factors Bax and Bak are recruited to the mitochondrial outer membrane, where they oligomerize and prompt the release of cytochrome c to the cytosol. After exiting the mitochondria, cytochrome c binds to apaf-1 and procaspase-9, forming the apoptosome and initiating the caspase signaling cascade that mediates irreversible apoptotic cell death. Initiated by a wide range of upstream apoptotic stimuli, the release of cytochrome c from the mitochondrion is facilitated by the remodeling of cristae, fusing together to form large intramitochondrial tubes to allow the large-scale release of inner membrane-associated cytochrome c out of the organelle [73]. This cristae remodeling is mediated by alterations in the interacting factors that regulate crista junction formation, particularly OPA1 and MiCOS; under steady-state conditions, OPA1 forms oligomers that help form crista junctions and prevent cytochrome c release [59,61]. Upon apoptotic induction, however, these OPA1 oligomers collapse, allowing multiple cristae to fuse together and promote cytochrome c release [53,74]. Accordingly, OMA1 plays a strong role in the mitochondrial apoptotic response by facilitating cytochrome c release. Moreover, OMA1 also interacts with MICOS to modulate contact sites between the inner and outer mitochondrial membranes [75], adding to an emerging complex set of protein interactions that mediate mitochondrial membrane structural dynamics, with direct impacts on apoptotic induction.

Following on from mitochondrial roles in apoptosis, mitochondrial dynamics have been shown to play critical roles in other signaling pathways, including autophagy and the integrated stress response (ISR). Under conditions including starvation and mTOR inhibition, cells will undergo autophagy to sequester and degrade a variety of cellular components, including mitochondria, as a key mechanism of cellular quality control. Mitochondrial autophagy (mitophagy) allows for the selective degradation of nonfunctional organelles. When Δψ_m_ is lost, mitochondrial pten-induced kinase 1 (PINK1) accumulates in the outer mitochondrial membrane, where it recruits and binds the parkin E3 ubiquitin ligase [76,77]. Following parkin recruitment, the ubiquitination of mitochondrial outer-membrane proteins targets the organelle via binding of the p62/SQSTM1 and the recruitment of microtubule-associated light chain 3 (LC3), which mediates the formation of an autophagosome around the organelle for subsequent degradation upon fusion with a lysosome [78]. Both mitochondrial fission and fusion factors are involved in this process. DRP1-mediated fission is required, but not sufficient, for mitophagy, as individual organelles must be separated into ‘bite-size’ organelles for autophagosome formation [78,79], while the inhibition of OMA1 activation similarly blunts mitophagy in cells actively undergoing mitochondrial ATP production [80].

More recently, mitochondrial stress has been shown to be communicated to the cell at large via the activation of the integrated stress response (ISR) via OMA1. Previously, genetically driven mitochondrial dysfunction was demonstrated to communicate with the nucleus [81], while more recently, the mitochondrial unfolded protein response has been shown to result in widespread changes in gene expression within the nucleus [82]. However, the mechanisms by which mitochondrial dysfunction is communicated to the nucleus (i.e., mitochondria–nucleus ‘retrograde’ signaling) remain unclear. Recently, OMA1 was shown to play a vital role in mitochondrial communication with nuclear gene expression. Distinct from its role in processing OPA1, the stress-induced activation of OMA1 leads to cleavage of the DELE1 inner membrane protein to its short form (S-DELE1). S-DELE1 then accumulates in the cytosol, binding and activating the HRI kinase. Active HRI phosphorylates e1F2a to both slow ribosomal protein synthesis and induce ISR in the nucleus through the activation of the transcription factors ATF4, ATF5, and CHOP [83,84]. These findings provide a direct mechanism by which mitochondrial dysfunction leads to profound changes in nuclear gene expression, distinct from autophagy and apoptosis.

Given the wide-ranging impacts of mitochondrial membrane-shaping dynamics, it is unsurprising that perturbations in these factors cause severe disease phenotypes in experimental models and clinical settings, acting as an underlying mechanism of mitochondrial involvement in human disease. As a bioenergetic organellar network with a dual genetic background, mitochondria were first shown to be a primary driver of human disease when pathogenic mtDNA mutations were first identified in 1988; a point mutation of mtDNA was shown to cause hereditary optic neuropathy [85], while large-scale deletions of mtDNA were found in patients with myopathy [86]. These seminal findings led to the discovery of an array of mutations in both mtDNA and nuclear genes that directly compromise oxidative phosphorylation as the classical mitochondrial neuromuscular diseases. Typically, normal and mutant mtDNAs coexist within the same cell and tissue, called heteroplasmy. Crucially, cells can typically withstand a relatively high mutation load (usually 60–80% of total mtDNA content) before oxidative phosphorylation is compromised; however, mtDNA mutations frequently display a sharp threshold effect, in which mutation loads above the threshold cause a collapse of mitochondrial ATP production. The precise pathogenic threshold level of heteroplasmy varies depending on the specific mutation; deletions tend to threshold at ~80%, while point mutations often have threshold values above 90% [8]. The genotype–phenotype relationships in classical mitochondrial neuromuscular disorders presaged findings of broader roles for mitochondrial dysfunction in prevalent human diseases. MtDNA deletions were found to occur at high, above-threshold levels in neural tissue of patients with Parkinson’s disease and aging [87,88], while deficits in mitochondrial bioenergetics have been subsequently found as part of the pathology of heart failure [89] and diabetes [90,91]. The emerging integration of OMA1-mediated mitochondrial stress sensing with autophagy and apoptosis increasingly shows key pathological impacts in a range of model systems and patient settings.

OPA1 was originally identified through its causative role in heritable optic atrophy [92]. OMA1’s role in activating OPA1 processing and apoptosis makes it a strong activator of diseases caused by cell death. The cardiac-specific activation of OMA1 and mitochondrial fragmentation causes heart failure [93], while OMA1 activation causes neuronal death in neurodegeneration [94] and under ischemic injury [95] in mouse models. Moreover, while alterations in mitochondrial metabolism drive changes in organellar dynamics, the converse is also true. OPA1 is required for metabolic shifting to oxidative phosphorylation during muscle differentiation via the supercomplex assembly factor SCAF1 [96], as part of a small but growing case for the importance of OPA1 in cellular development [97,98,99]. The ablation of OMA1 causes disrupted metabolism and obesity in mice [66]. As OMA1 is pro-apoptotic through OPA1 processing, the inhibition of OMA1 promotes glycolytic metabolism and tumor progression in colorectal cancer [100], while activating OMA1 increases chemotherapeutic effectiveness in ovarian cancer settings [101]. Similarly, OPA1 deletion confers protection from liver injury and drug toxicity [102]. Taken together, these findings demonstrate a crucial role for OMA1 and OPA1 as a central stress-sensitive pathway for cellular homeostasis across a wide range of prevalent human diseases, while the modulation of OMA1 and OPA1 provides a promising avenue for translational approaches.

## 5. Interactions, Domains, and Physiology within the Organellar Network

Moving forward, a more integrated understanding of mitochondrial biology will incorporate the higher-order organization of mitochondrial structural dynamics, metabolism and bioenergetics, and cellular signaling functions, while also exploring the adaptive heterogeneity within the mitochondrial network. Mitochondria have long been appreciated to be highly protein-rich, as was originally apparent from their high contrast in transmission electron microscopy [1,2]. Within the organelle, the mitochondrial inner membrane is extremely protein-rich due to the presence of multiple complexes of oxidative phosphorylation, as well as mitochondrial protein import machinery and other crucial metabolic and proteolytic factors. Significantly, the OxPhos complexes interact with each other in macromolecular supercomplexes to facilitate electron transfer and metabolism [59,103]. More recently, additional factors appear to be associated in localized domains of interacting proteins with important functional relevance. ER–mitochondrial contact sites contain mitochondrial fission and fusion factors, allowing responsive membrane dynamics at the site of mitochondrial–ER interaction [104], while MICOS links cristal organization with ER–mitochondria contact sites [105,106]. Similarly, the higher-order organization of the OMA1 stress-sensing machinery appears to involve microdomains within the inner membrane, often in association with MICOS or the mitochondrial scaffolding proteins prohibitin 1 and 2 (Figure 2). The scaffolding protein stomatin-like protein-2 (SLP-2) localizes to the matrix face of the inner membrane, where it interacts with YME1L and prohibitin to form the SPY complex [107]. SLP-2 also interacts with the mitochondrial ribosome to promote the synthesis of mtDNA-encoded proteins [108]. This complex has several impacts on OPA1. YME1L constitutively cleaves L-OPA1, acting together with OMA1 [64], while SLP-2 stabilizes L-OPA1 by inhibiting OMA1 [109]. OMA1 is likely maintained as a multimeric assembly within the inner membrane [110] and interacts with both prohibitin [69] and MICOS independent of OPA1 [75]. OPA1 associates with MICOS and SLC25 to carry out its role in the formation of cristal junctions [61] (Figure 2). In addition to this complex series of identified interactions between these factors, the interacting factors in situ within the inner membrane likely involve multimeric arrangements. In *S. cerevisiae*, mitochondrial prohibitins appear to form multimeric ring complexes [111], while OMA1 appears to similarly exist as an oligomeric complex [110]. Taken together, these findings suggest that macromolecular domains within the organellar network allow for the efficient compartmentalization and distribution of key mitochondrial functionalities. The precise arrangement and organization of these functional microdomains within the inner membrane remain to determined but are likely to be of strong functional importance. For example, the N-terminal unstructured region of OMA1 is exposed on the matrix face of the inner membrane and is required for sensing the loss of Δψ_m_ [65]; it is easy to envision how a variety of regulatory interactions with this matrix-associated OMA1 region could be determined by the domain organization of OMA1 with its interacting partners, as well as with adjacent domains such as the SPY complex. These complex macromolecular interactions likely involve the lipid components of the inner membrane as well; cardiolipin is increasingly revealed to play an important role in interacting with OPA1 [46,112], as well as interacting with OMA1, regulating metalloprotease in concert with prohibitin [69]. Moreover, mitochondrial organization and dynamics are increasingly revealed to be highly fluid. Advances in super-resolution microscopy demonstrate that mitochondrial cristae undergo continuous remodeling [113], while the mitochondrial matrix itself undergoes changes in viscosity, with metabolic stress causing macromolecular crowding and decreased matrix fluidity [114]. While it is convenient to describe Δψ_m_ as either ‘intact’ or ‘dissipated’, Δψ_m_ is heterogenous within the organelle, with distinct values between the cristal membrane versus the inner boundary membrane, as well as between individual cristae [58]. OMA1 protects against the hyperfusion of mitochondria, activated by transient ‘flickering’ depolarizations of Δψ_m_ within individual organellar foci [115].

Taken together, these findings indicate that macromolecular domains of interacting factors within the organellar network provide the distribution of specific mitochondrial functionalities, with key impacts on the highly fluid dynamics of cellular physiology, both within discrete regions of the organelle and across the cell as a whole.

## Figures and Tables

**Figure 1 ijms-25-04566-f001:**
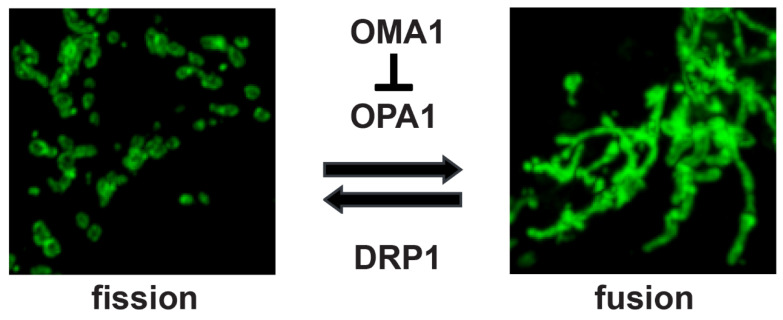
**Mitochondria balance between fission and fusion.** Mitochondrial fission, or division, is carried out by the recruitment of the cytosolic GTPase DRP1. Conversely, the fusion of the inner membrane is carried out by OPA1. Under cellular stress, OMA1 metalloprotease cleaves fusion-active OPA1, leaving fission unopposed. SHSY5Y cells visualized by anti-TOM20 immunolabeling, Nikon AX confocal.

**Figure 2 ijms-25-04566-f002:**
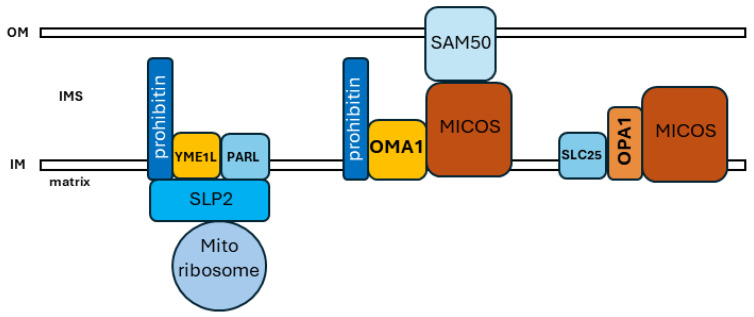
**Schematic of OMA1 and OPA1 macromolecular associations within the inner membrane.** In addition to the regulated maintenance of oxidative phosphorylation complexes into supercomplexes and dimeric assemblies of the F_1_F_0_ ATP synthase, increasing evidence indicates that other critical mitochondrial functionalities may be carried out by domains of interacting factors within the inner membrane. Prohibitin-associated SLP2 interacts with both the mitochondrial ribosome and inner-membrane proteases PARL and YME1L as part of the SPY complex (**left**), while OMA1 interacts with the MICOS complex as part of contact sites between the outer and inner membrane (**center**). Independently, OPA1 associates with MICOS and SLC25 to mediate crista junction formation (**right**).

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
