# Peer review of "OMA1-Mediated Mitochondrial Dynamics Balance Organellar Homeostasis Upstream of Cellular Stress Responses"

_ijms, 2024, doi:10.3390/ijms25084566_

Round 1

Reviewer 1 Report (Previous Reviewer 2)

Comments and Suggestions for Authors

Thanks for the response from the authors. As I mentioned, this review is very similar to other mitochondria dynamics review except for the "OMA1" part. The review will be more unique and valuable if you focus on OMA1. It would be helpful if you can point out the novelty of your review. I agree the mitochondria dynamics is important for the cellular differentiation and development; however, the ideas were already published. This review does not provide new ideas compared with the published reviews, this is the reason I don't think this review can be beneficial for this field. If this review can highlight the OMA1 part and the related significance would be better and considered to be published.

Author Response

We thank the reviewer for their comments. We have amended the text to focus the reader's attention on OMA1, per the reviewer's request. In truth, the manuscript was already somewhat 'OMA1-centric', but the new title and edits focus more on OMA1's unique role as a stress-sensitive mediator of mitochondrial structure/function balance, with profound effects on the cell at large.

Reviewer 2 Report (Previous Reviewer 1)

Comments and Suggestions for Authors

Authors have satisfactorily addressed the comments.

Comments on the Quality of English Language

Minor editing of English language required.

Author Response

We thank the reviewer for their time and consideration in carefully evaluating our manuscript.

Round 2

Reviewer 1 Report (Previous Reviewer 2)

Comments and Suggestions for Authors

Thanks for accepting my opinion about OMA1-centric idea. Since the overall structure is still very similar to other reviews, it would be better to have a paragraph for OMA1 roles in mitochondria dynamics and physiology.  

Author Response

We thank the reviewer for this comment. We have amended the manuscript to include a paragraph delineating more specifically the impact of OMA1 on mitochondrial dynamics and physiology, including a discussion of the studies examining OMA1 knockout and its effects on mitochondrial metabolism.

This manuscript is a resubmission of an earlier submission. The following is a list of the peer review reports and author responses from that submission.

Round 1

Reviewer 1 Report

Comments and Suggestions for Authors

In this review, authors have cited the role of mitochondrial dynamics in regulating cellular state and initiating integrated stress response. Overall manuscript is well written and is of interest to the readers. Authors can consider following points to add on.

Recent studies have shown crucial role of cardiolipin in regulating mitochondrial dynamics, bioenergetics and apoptosis. CL regulates both division and inner membrane fusion including OXPHOS regulation. This can be elaborated in detail.

Does integrated stress response in relation to ER stress or amino acid starvation regulates mitochondrial dynamics differently?

What about role of mitochondrial dynamics on cell cycle or cell proliferation?

Significance of PGC1alpha and PPARalpha as principal regulators of mitochondrial fusion-fission dynamics.

Comments on the Quality of English Language

Minor editing of English language required.

Reviewer 2 Report

Comments and Suggestions for Authors

This review described the concepts of how mitochondria change their shape and the significant effects of the change on cellular events, similar to many articles I have read. The content about mitochondrial dynamics is not that different from other articles I read before, except for the part of "OMA1". However I checked the reference and I found the author already published the review article about OMA1. Thus I don't think this review can be too beneficial for this field.  

Comments on the Quality of English Language

The writing style is clear and understandable. Here I just list one sentences (words) I thought ambiguous.

When the transmembrane potential across the inner membrane (Dym) is lost: What does the Dym mean?